# Design of multi-modal antenna arrays for microwave hyperthermia and $^1$H/$^{19}$F MRI monitoring of drug release

**Daniel Hernandez**[1], **Taewoo Nam**[2], **Eunwoo Lee**[2], **Jae Jun Lee**[3], **Kisoo Kim**[4]*, **Kyoung Nam Kim**[5]*

1 Neuroscience Research Institute, Gachon University, South Korea, 2 Department of Health Sciences and Technology, GAIHST, Gachon University, South Korea, 3 Non-Clinical Center, KBIO Osong Medical Innovation Foundation, Cheongju-si, Chungbuk, Korea, 4 Department of Biomedical Engineering, Kyung Hee University, Yongin, South Korea, 5 Department of Biomedical Engineering, Gachon University, Seongnam, South Korea

* kisoo.kim@khu.ac.kr (KK); kyoungnam.kim@gachon.ac.kr (KNK)

**Data Availability Statement:** All relevant data are available from the Figure database (10.6084/m9. figshare.26827867).

## Abstract

This simulation-based study presented a novel hybrid RF antenna array designed for neck cancer treatment within a 7T MRI system. The proposed design aimed to provide microwave hyperthermia to release $^{19}$F-labeled anticancer drugs from thermosensitive liposomes, facilitating drug concentration monitoring through $^{19}$F imaging and enabling $^1$H anatomical imaging and MR thermometry for temperature control. The design featured a bidirectional microstrip for generating the magnetic |B$_1$|-fields required for $^1$H and $^{19}$F MR imaging, along with a patch antenna for localized RF heating. The bidirectional microstrip was operated at 300 MHz and 280 MHz through the placement of excitation ports at the ends of the antenna and an asymmetric structure along the antenna. Additionally, a patch antenna was positioned at the center. Based on this setup, an array of six antennas was designed. Simulation results using a tissue-mimicking simulation model confirmed the intensity and uniformity of |B$_1$|-fields for both $^{19}$F and $^1$H nuclei, demonstrating the suitability of the design for clinical imaging. RF heating from the patch antennas was effectively localized at the center of the cancer model. In simulations with a human model, average |B$_1$|-fields were 0.21 µT for $^{19}$F and 0.12 µT for $^1$H, with normalized-absolute-average-deviation values of 81.75% and 87.74%, respectively. Hyperthermia treatment was applied at 120 W for 600 s, achieving an average temperature of 40.22°C in the cancer model with a perfusion rate of 1 ml/min/kg. This study demonstrated the potential of a hybrid antenna array for integrating $^1$H MR, $^{19}$F drug monitoring, and hyperthermia.

## I. Introduction

Head and neck cancer is one of the most commonly diagnosed cancers with increasing incidence and mortality rates [1]. Effective treatment is crucial for patient survival. The current treatment options include surgery, radiation therapy, and chemotherapy. Although

**Funding:** This study was partially supported by National Institutes of Dental and Craniofacial Research (NIDCR), USA under Grant K99DE032397, and supported by a grant from the Osong Medical innovation Foundation R and D Project funded by the Republic of Korea's Ministry of Health and Welfare (No: HI22C1989).

**Competing interests:** The authors have declared that no competing interests exist.

chemotherapy is effective, systemic treatment can cause severe toxic side effects on normal cells and organs. To address these challenges, targeted drug delivery technologies have been developed to enhance treatment efficacy by increasing the concentration of drugs at the tumor site, while minimizing exposure to healthy tissues. These technologies achieve this by targeting drugs to specific cells, releasing drugs in response to certain conditions, or using heat therapy to promote drug release from liposomes [2–6].

Clinical hyperthermia (HT) has been employed as a complementary cancer treatment to sensitize tumors to radiotherapy and chemotherapy by elevating the target tissue temperature to 40–45°C for 30–60 min [7–9]. In the past few decades, HT has received much interest in liposomal drug therapy to enhance the effectiveness of targeted drug delivery as an external trigger. Temperature-sensitive liposomes (TSL) [2] have been demonstrated as a carrier of anti-cancer drugs in clinical HT treatments [3]. In general, TSL is composed of dipalmitoyl-phosphatidyl-choline (DPPC) with the gel-to-liquid crystalline phase transition temperature (40–41°C) [10]. The bilayer of the TSL is not stable at mild HT temperature and finally releases the entrapped drug within a target region. Currently, doxorubicin encapsulated in the TSL, i.e., ThermoDox® (Celsion Corporation, Lawrenceville, NJ, USA), has been shown in clinical trials in combination with radiofrequency ablation and high-intensity focused ultrasound treatments [11, 12].

HT delivery can be achieved through various methods, such as radiofrequency, microwave, and ultrasound systems, to deliver targeted and localized HT to tumor targets [13, 14]. These techniques are often combined with magnetic resonance imaging (MRI) for image-guided therapy control and patient safety, as MRI offers superior soft-tissue contrast and non-invasive temperature information during HT procedures. The Pyrexar BSD-2000 MR-compatible applicator was designed for deep HT to deliver radiofrequency energy to tumors in the brain, prostate, or liver [15, 16]. This MR-guided RF applicator allowed planning, delivery, and control of deep HT with reproducible RF heating. For local HT, MR-guided focused ultrasound ablation systems have been investigated for the use of ultrasound HT and enhanced drug delivery [17, 18]. Several studies have demonstrated the potential benefits of MR-guided focused ultrasound-based HT delivery to rectal and head/neck cancers [19–21].

MRI has been shown to be capable of precise temperature monitoring for controlled drug delivery and monitoring of drug accumulation and release [4, 17, 22]. Several MRI contrast agents encapsulated in TSL have been investigated to monitor liposome distributions and concentrations in MRI. Dewhirst et al. demonstrated that MnSO4-loaded liposomes could be used for in-vivo monitoring of liposome concentration and distribution using MR T1 imaging [23]. Gadolinium-loaded liposomes have also been used to measure the concentration of drugs entrapped in liposomes [24]. When these paramagnetic agents are released, susceptibility effects induce the evolution of the intrinsic relaxation rate (1/T1), which can be used to estimate the drug concentration.

However, methods based on paramagnetic MR contrast agents for drug release monitoring have fundamental limitations when applying MR thermometry during HT procedure. Because MR thermometry methods are vulnerable to susceptibility changes leading to temperature errors, a reliable method for monitoring the temperature of the target tissue during the HT procedure is lacking. Hence, TSL drugs and MR contrast agents are usually injected after HT treatment [25, 26]. Methods for the simultaneous MRI monitoring of temperature and drug release should be explored and developed to achieve faster and more precise monitoring of drug release during treatment.

[19]F MRI is a non-invasive and non-toxic technique that can track [19]F-labeled compounds in drugs. Since the resonance frequency of [19]F (40.05 MHz/Tesla) is different from the proton

resonance frequency (42.58 MHz/Tesla), it enables to monitor [19]F-labeled drugs, together with the use of [1]H MR imaging. This approach may have the potential to optimize drug delivery and improve the efficacy of therapeutics. Langereis et al. [27] first introduced a temperature-sensitive liposomal with [1]H chemical exchange saturation transfer (CEST)- and [19]F-based contrast agents for MR-guided drug delivery. This liposome allows for simultaneous monitoring of drug distributions with [1]H CEST MR imaging and quantification of released drugs with [19]F MR imaging. Despite the potential benefits, there has been little further development of [19]F MRI and HT-mediated drug delivery since there is a lack of integrated drug delivery systems in combination with HT devices and [1]H/[19]F MRI dual RF coils.

Apart from [19]F MRI, MRI-compatible HT systems with integrated RF coils have been developed to achieve a higher resolution, signal-to-noise ratio (SNR), and accurate MR thermometry imaging. Winter et al. developed a bow tie antenna application that can provide RF heating and [1]H MR imaging in one device with 8-channel antennas operating at 7.0 Tesla MRI [28]. Recently, an MR-compatible head and neck HT applicator (MRcollar) was developed to perform RF heating and [1]H MR imaging in one device [29]. This applicator consisted of 12 antennas for heating (434 MHz) and 8-channel MR coils operating at 1.5 Tesla MRI. Despite recent advances in HT applicators for drug delivery with HT-mediated TSL, there are no reliable devices and methods for monitoring the temperature and drug release within the target tissue during the HT procedure.

For MRI, it is essential that coils or antennas produce uniform and strong magnetic $|B_1|$-fields, which are crucial for both signal transmission and reception [30]. These $|B_1|$-fields must operate at frequencies corresponding to the Larmor frequency of each nucleus being imaged. The Larmor frequency is determined by the strength of the main magnetic field ($B_0$) of the MRI scanner. For instance, in a 7T MRI scanner, the Larmor frequency for [1]H is approximately 300 MHz, while for [19]F it is around 280 MHz. Ensuring that the coils or antennas produce $|B_1|$-fields at these specific frequencies is vital for obtaining high-quality images and accurate data during MRI procedures.

In MRI, the primary characteristic of the antenna is its ability to generate the $|B_1|$-field. It is also desirable for the electric $|E|$-field to be minimal to reduce the specific absorption rate (SAR), a key safety concern in MRI. However, for the development of thermal applicators, the focus has shifted to enhancing $|E|$-field generation. A higher $|E|$-field interacts with tissues, and through Ohmic resistance, it absorbs energy that can raise tissue temperature. Consequently, patch antennas, dipole antennas, and bowtie antennas are commonly used for non-invasive temperature generation in medical applications due to their effective $|E|$-field generation capabilities. With the $|E|$-field, temperature maps can be calculated based on the Pennes bioheat method.

The aim of this study was to develop a novel hybrid antenna array that integrates three key functions: microwave HT, and [19]F and [1]H imaging. We introduced the concept of a bidirectional microstrip antenna that created asymmetric current pathways along the conductor line to produce two different frequencies from the excitation ports at both ends of the antenna. To ensure compatibility with MRI and human size scales, the bidirectional microstrip antenna was modified to reduce the size by implementing the conductor line in a loop shape while maintaining asymmetrical current pathways between the ports. This design also included a centrally located patch antenna within the loop microstrip to deliver the targeted heat for HT. The proposed microstrip and patch antenna arrangements were then extended to an array setup. Electromagnetic and thermal simulations were performed to calculate $|B_1|$-fields, $|E|$-fields, and temperature maps for a neck cancer model, demonstrating the feasibility of the proposed antenna array for clinical translation.

## II. Material and methods

### A. Electromagnetic simulations

We conducted electromagnetic (EM) simulations using finite element analysis in a commercial software package (Sim4Life v7.2.1.11125, Zurich Med Tech AG, Switzerland). During the simulation, we calculated the impedance, S-parameters, magnetic $|B_1|$-fields, electric $|E|$-fields, SAR, and temperature maps. All conductors were modeled as perfect electrical conductors (PEC). The excitation signal was set as a Gaussian pulse with specified central and bandwidth frequencies (see the details in the following section). The $|E|$-fields obtained from the EM simulation were used as a heat source in the Pennes bioheat model to calculate the three-dimensional temperature distributions.

### B. Fundaments of the bidirectional microstrip

In a typical microstrip configuration, the conductor structures are symmetrical, resulting in consistent electromagnetic patterns regardless of the direction of the current. However, a recent study introduced the concept of a dual-port, dual-band microstrip for satellite communications, capable of delivering different frequencies depending on the excitation port [31]. Based on this concept, we designed a bidirectional microstrip with asymmetrical structures for MRI applications. This microstrip could resonate at two different frequencies, as determined by the excitation port used. The microstrip lines were designed to be asymmetrical according to the port directions, allowing for different resonance frequencies depending on the direction of excitation.

To demonstrate the proposed concept, we first examined the performance of the microstrip by incorporating an asymmetric conductor. The linear microstrip consisted of a conducting line 285.21 mm in length and 20.53 mm in width, positioned on a dielectric material with a height of 10 mm and a permittivity of 4, based on FR-4. Fig 1 shows the design of the simple bidirectional microstrip used for the proof of concept, with the input ports located at each extreme (Port #1 and Port #2). To illustrate bidirectional resonance, a small conductor, referred to as a "wing" in this study, was added perpendicular to the conductor line. The wing measured 40 mm in length along the X-axis and 5 mm in thickness. Its position was varied from one extreme to the other, from -Z to +Z, in 15 steps of 20 mm each as indicated with the dotted line along the microstrip in Fig 1. The S-parameter (S11) was used as a measure of the resonance frequency of the microstrip. Each port was excited with a Gaussian pulse with a central frequency of 300 MHz and a bandwidth of 600 MHz.

### C. Integrated bidirectional microstrip and patch antenna

The size of the microstrip antenna should be reduced to adapt the bidirectional microstrip concept for MRI applications. To achieve this, we applied a microstrip surface coil design that has been shown to minimize the size of the antenna [32]. The length of the conductor line of the microstrip loop coil is proportional to the wavelength of the operational frequency and the dielectric material [32] that it is mount on. The size of the patch antenna was calculated using the common microstrip antenna equation [33]. This approach facilitates the use of microstrip coils suitable for placement alongside therapeutic devices within the restricted space of the MRI bore. As shown in Fig 2, a dielectric material of 125 mm x 125 mm x 10 mm was positioned, and the microstrip line on the top of the dielectric was configured to match the perimeter of the rectangular surface, with a width of 5 mm. A ground plate was placed at the bottom of the dielectric material. The power ports 1 and 2 operate at frequencies of 300 MHz and 280 MHz, respectively (Fig 2A). These frequencies were tuned using two extra lines (W1 and W2

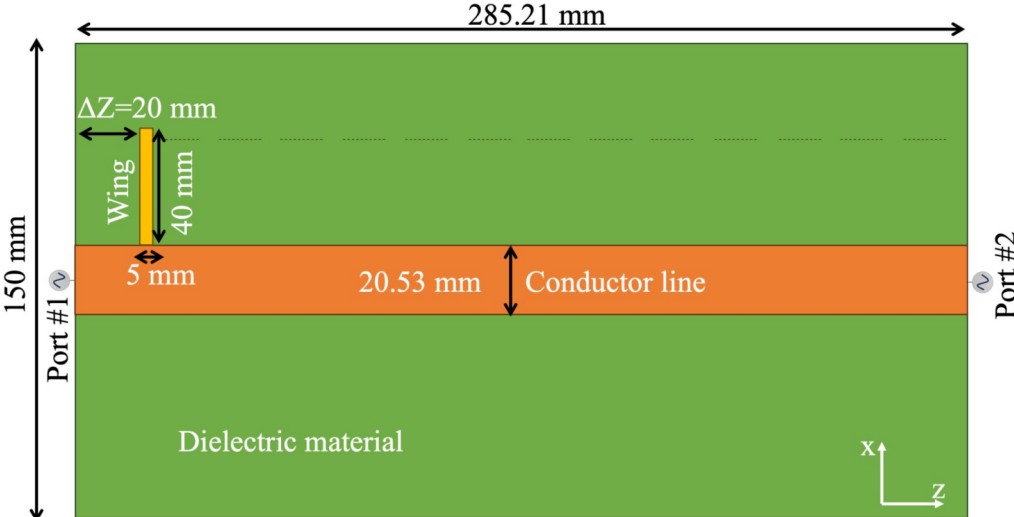

**Fig 1. A planar microstrip was used for demonstration purposes.** This bidirectional microstrip features dual-port excitations and a wing line (yellow) with variable positions along the microstrip. To create an asymmetric conductor line, the wing line is shifted to different positions with a step size of $\Delta Z = 20$ mm.

shown in Fig 2A) with lengths of 50 mm and 10 mm. The sizes of W1 and W2 were determined by analyzing the effects of their length and resonance frequency through iterative simulations. An analysis of this parameter is presented in the results section.

Applications utilizing patch antennas have been investigated for use in HT treatment [34, 35]. These antennas are uniquely tailored to generate electric fields (E-fields). The

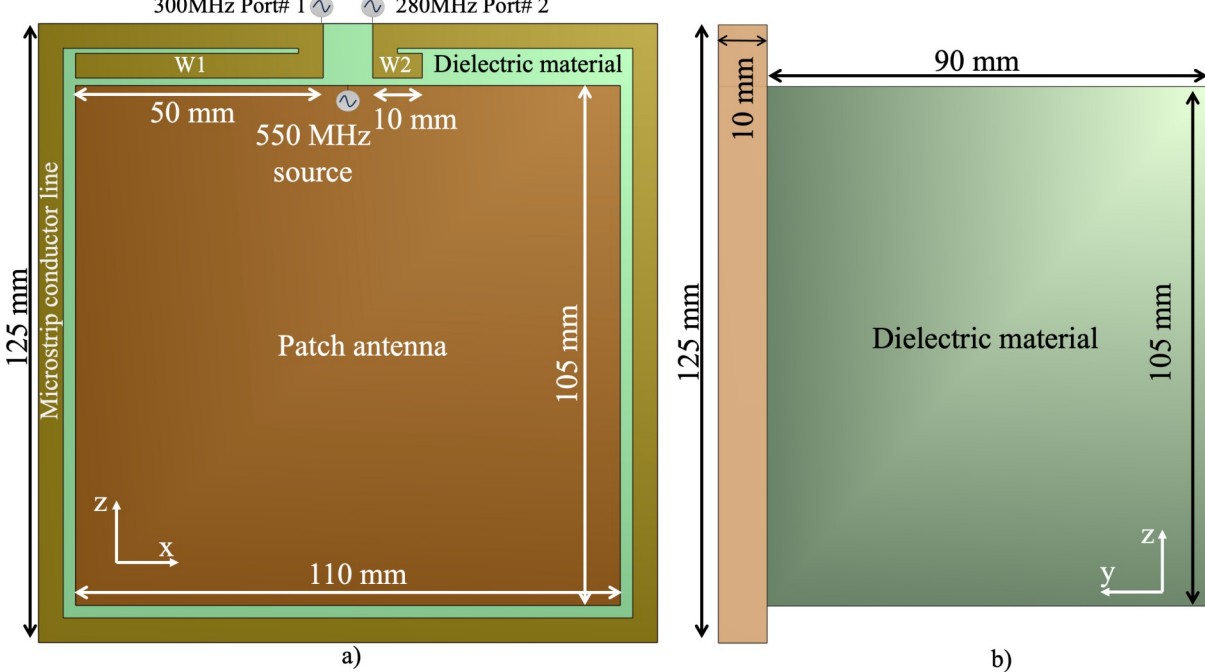

**Fig 2.** The geometry of a single hybrid antenna with the bidirectional microstrip surface coil and patch antenna in a) the top view showing the excitation ports and the asymmetric wing lines, and b) the lateral view showing the dimensions of the dielectric materials.

operational frequency of a patch antenna is modulated through its geometry and materials. It comprises a ground plane, a dielectric material situated between two conductor planes, and an electric current source positioned between these planes. The operational frequency of the antenna is determined by the dimensions of the conductor and dielectric, coupled with their permittivity [36]. In this study, a patch antenna with dimensions of 110 mm × 105 mm was placed at the center of the module (Fig 2A). The source of the patch antenna was placed on the top area. The geometry and dielectric materials of the patch antenna were designed to enable operation in the frequency range of HT applications, such as 550 MHz. An additional dielectric material was placed beneath the microstrip dielectric material. As depicted in Fig 2B, the lateral view of the coil configuration shows a dielectric material with dimensions of 110 mm x 105 mm x 90 mm, enabling tuning to a frequency of 550 MHz. In the simulation, the electrical properties of the dielectric material were set to represent FR4, with a permittivity of 4. The ground plane was placed to fully cover the dielectric material.

## D. Simulation model and hybrid antenna array

Based on the design of the integrated microstrip and patch antenna, six antennas were assembled in an array with a diameter of 280 mm, featuring a 60˚ angular separation between antennas (Fig 3). To evaluate the distribution of magnetic $|B_1|$ and E-fields, a simulation model mimicking muscle tissue with a cylindrical shape with a diameter of 140 mm and a length of 150 mm was used. A sphere of 20 mm in diameter was also added to represent the cancer tissue. The electrical properties of the simulation model and the mimicking cancer tissue are summarized in Table 1.

To avoid unwanted heating on the surface, a water coupling pad (matching medium) was simulated and located surrounding the simulation model. The water pad was cylindrical, with dimensions of 260 mm in diameter and 150 mm in height. The electrical properties of the water pad were selected with a permittivity of 76.70 and conductivity of 5 x $10^{-5}$ S/m. Three different locations of cancer were used to evaluate the hybrid antenna array with the six modules (Fig 3). We performed a steady-state thermal simulation to observe hyperthermic temperature distributions. The patch antennas were operated at 550 MHz with a normalized total power of 90 W (Fig 3A), 57 W (Fig 3B), and 48 W (Fig 3C) for 600 s. These varying power levels resulted in similar or identical steady-state temperature increases, reaching a maximum temperature of approximately 43˚C within the cancer model. The initial tissue temperature was set to 37˚C for the muscle and cancer simulation models, and 18˚C for the water pad. The water pad, muscle, and cancer simulation models were set with thermal conductivity of 0.6045, 0.49, and 0.50 W/m/K, and heat capacity of 4178, 3421, and 3415 J/kg/K, respectively.

## E. Simulation-based study with a human model

To demonstrate the feasibility of the hybrid antenna array for clinical translation, a human model ([DUKE model in Sim4Life software] [37]) was used with a spherical cancer model. The cancer tissue was located inside the neck muscle tissue with a radius of 20 mm. The electrical and thermal properties of the tissues in the human model were provided by the database in the software based on previous measurements and were updated according to the frequency of operation [38]. A water pad was added to prevent temperature elevation in the skin. The water pad was cylindrical, with dimensions of 260 mm in diameter and 150 mm in height. Fig 4 shows the positioning of the hybrid antenna array in the head and neck model with the cancer tissue. The array consists of six antennas placed around a circumference of 280 mm in diameter with a 60˚ angular separation between antennas. Similar to the thermal simulation with the simulation model, the steady-state HT was observed by the patch antennas, operated at 550

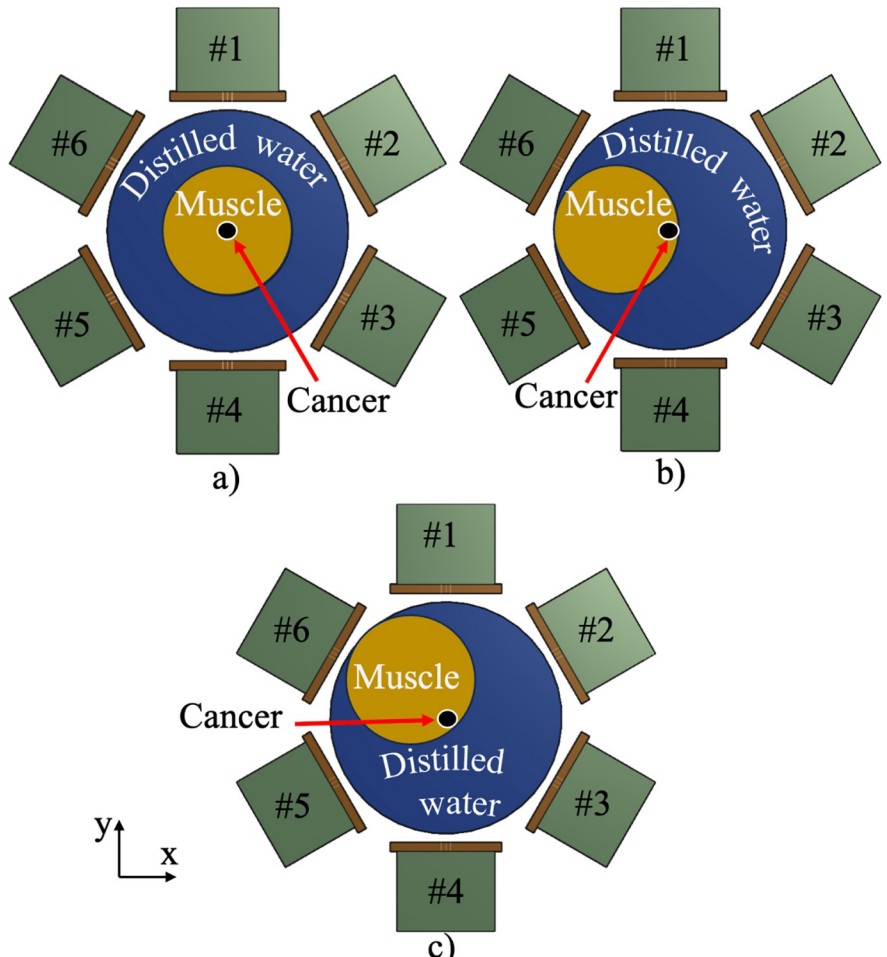

**Fig 3. The hybrid antenna array consists of six elements, along with the water pad, muscle, and cancer simulation models.** Different locations of the muscle and cancer simulation models were tested: a) in the central position, b) with the muscle simulation model shifted 60 mm to the left, and c) shifted 42.5 mm to the left and upward.

MHz with a normalized total power of 225 W for a duration of 600 s. The initial tissue temperature was set to 37˚C for the muscle and cancer simulation models and 18˚C for the water pad. The thermal properties of the water pad and the cancer/surrounding tissues were defined with thermal conductivities of 0.6045 and 0.49 W/m/K, heat capacities of 4178 and 3421 J/kg/K, and perfusion rates of 3 ml/min/kg, respectively [39]. Furthermore, we examined a lower perfusion rate in the cancer model than in the surrounding tissue to explore different clinical scenarios. A perfusion rate of 1 ml/min/kg was assigned to the cancer model and 3 ml/min/kg to

**Table 1. Electrical properties of the muscle and cancer-mimicking tissues at each frequency.**

|  | Frequency [MHz] | | |
| --- | --- | --- | --- |
| **Tissue Type** | **280** | **300** | **550** |
| Muscle permittivity | 58.45 | 58.2 | 56.18 |
| Muscle conductivity [S/m] | 0.77 | 0.77 | 0.84 |
| Cancer permittivity | 64.34 | 64.02 | 61.80 |
| Cancer conductivity [S/m] | 0.84 | 0.85 | 0.92 |

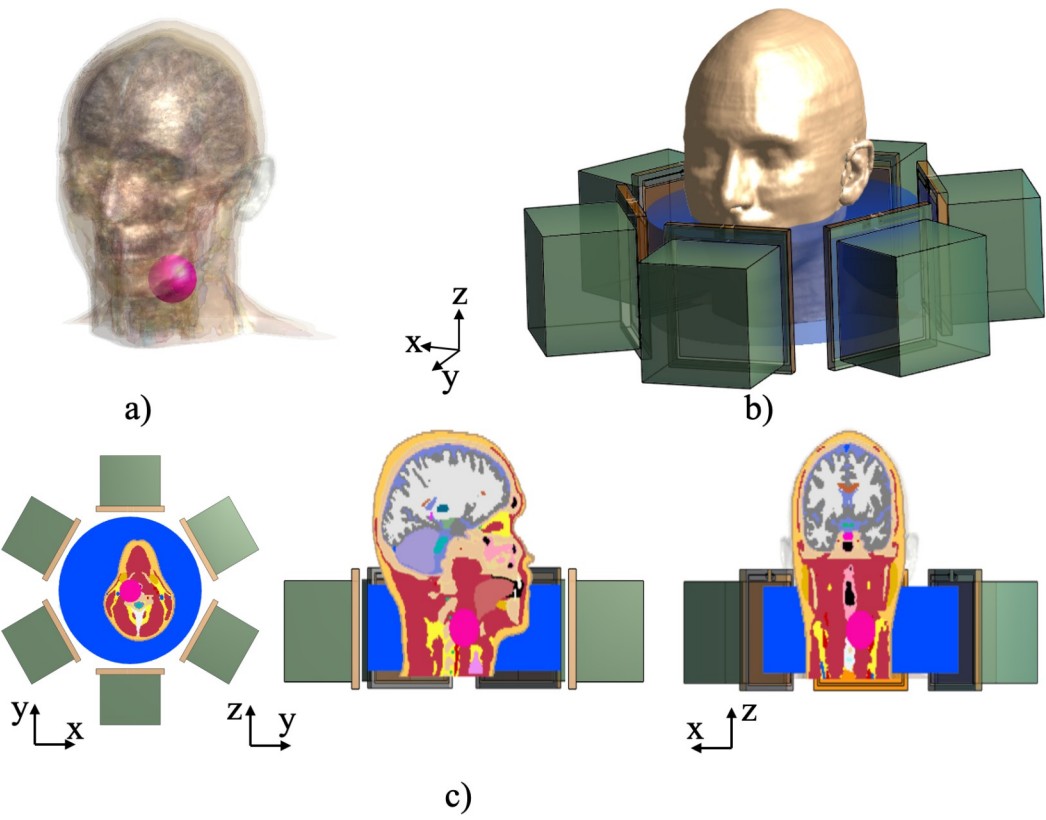

**Fig 4.** a) The human head and neck model with the cancer tissue shown as a pink sphere. b) The positioning of the array applicator around the neck. c) The X-Y view showing the position of the cancer inside the neck model.

the surrounding tissue. A normalized total power of 120 W was set to achieve steady-state temperature increase, reaching a maximum temperature of 43°C within the cancer model for 600 s.

## F. Evaluation factors

To evaluate the performance of the hybrid antenna array, three evaluation factors were calculated as follows:

1. **Uniformity Analysis (NAAD):** The uniformity of the $|B_1|$-field was assessed using the normalized absolute average deviation (NAAD).

$$NAAD = 100 \times \left( 1 - \frac{1}{N \times \bar{I}} \sum_{i=1}^{N} |I_i - \bar{I}| \right) \tag{1}$$

where the number of pixels, $|B_1|$-intensity of each pixel, and the average of $|B_1|$-intensity were expressed as $N$, $I_i$ and $\bar{I}$, respectively.

2. **Specific absorption rate (SAR):** The input power was normalized to 1 W when calculating the 10 g-averaged SAR ($SAR_{10g}$).

$$SAR_{10g} = \frac{1}{M} \int_{R(M)} \frac{\sigma E^2}{2\rho} dm \tag{2}$$

where the region mass (M), conductivity (σ), which is provided by the simulation software,

E-field (E), and mass density (ρ). The electric fields produced by the antennas were used for computing the SAR, which was performed using a simulation software.

3. **Temperature estimation:** Temperature maps were computed using the Pennes Bioheat Equation [40] in the simulation software. This equation describes the change of temperature $T$ [˚C] in in the tissue over time $t$ as follows:

$$\rho C \frac{dT}{dt} = \nabla[k \cdot \nabla T] - \omega_b C_b [T - T_b] + Q_{RF} \tag{3}$$

Where $Q_{RF}$ is the RF energy deposition in the tissue [W/m$^3$], $C_b$ is specific heat of blood [J/ ˚C/kg], $C$ is the specific heat capacity [J/˚C/kg], $T_b$ is blood temperature [˚C], $k$ is thermal conductivity [W/m/˚C], $\omega_b$ is blood perfusion [kg/m$^3$/s], and $\rho$ is the density of tissue [kg/ m$^3$]. To analyze the temperature distribution of HT in a specific area, we used the T10, T50, and T90 metrics [41]. The T10, T50, and T90 metrics indicate the temperatures reached or exceeded by 10%, 50%, and 90% of all voxels within the cancer model during the heating period, respectively.

## III. Results

### A. Test for the concept of bidirectional microstrip

To demonstrate the concept of bidirectional resonance of the microstrip, electromagnetic simulations were performed with the microstrip line shown in Fig 1, in which the wing conductor line was shifted in steps of 15 mm along the conductor line. For each position of the wing, the impedance and S11 parameters were measured for each port. As shown in Fig 5, the effect of moving the wing to different positions along the conductor line is plotted, where the frequency of the maximum impedance shifts according to the position of the wing. The black line represents the situation in which the wing is placed at the center (140 mm from port 1) in symmetry, resulting in a frequency of 292 MHz (Fig 5). In contrast, when the wing was positioned at distances of 20 mm and 260 mm from port 1, the frequency was 276 MHz. Similarly, when the wing was located 80 mm and 200 mm away from port 1, the frequency was measured at 285 MHz. This plot indicates the effects of the wing position on the impedance and frequency of the microstrip.

The S11 parameter was plotted according to the position of the wing at ports 1 and 2. In Fig 5B, the black and blue lines represent the S11 measured at ports 1 and 2, respectively. The resonance frequency could be affected by factors such as the size of the wing and the angle between the conductor line and the wing. In Fig 5C, the frequency difference from each port at the position of the wing is plotted, with a maximum difference of 19 MHz when the wing is placed 40 or 240 mm from excitation port 1. It should be noted that when the wing is placed at the center, there is zero frequency difference.

Fig 5D and 5E show the |B$_1$|-fields of the microstrip when the wing is placed 40 mm from port 1, where the tuning frequencies were 344 MHz and 325 MHz at ports 1 and 2, respectively. These plots demonstrate that two different frequencies can be obtained from the perspective of each port by placing a wing along the line.

### B. Single module with integrated microstrip and patch antenna

The EM simulations were conducted to verify the resonance frequency of a single module. The initial analysis involved examining the impact of varying the lengths of W1 and W2. Simulations were performed by changing the lengths of W1 and W2 from 0 to 50 mm in six steps of

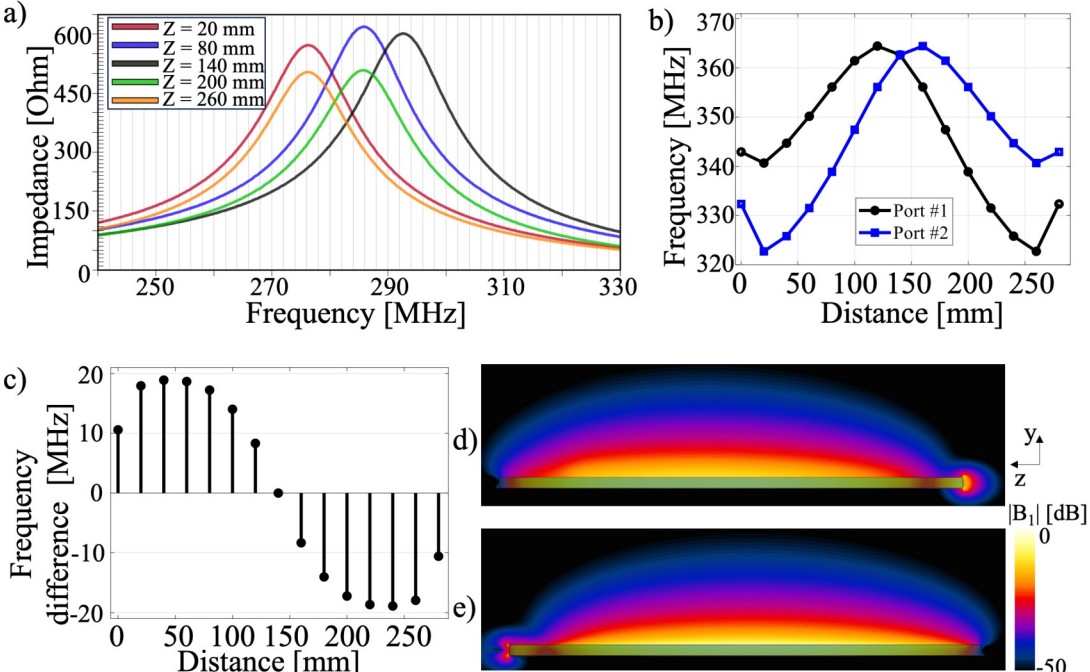

**Fig 5.** The plots illustrate the following relationships: a) The change in impedance versus frequency as a function of the wing position. b) The resonance frequency as a function of the wing distance, viewed from ports 1 and 2. c) The frequency difference between ports 1 and 2 at each wing position. d) The $|B_1|$-field produced by the microstrip when the wing is positioned 40 mm from port 1, showing the field at 344 MHz. e) The $|B_1|$-field produced by the microstrip when the wing is positioned 40 mm from port 1, showing the field at 325 MHz at port 2.

10 mm, as indicated by the black dots in Fig 6A. A linear fit for the frequencies $f$ and $W$ was determined as follows:

$$f[MHz] = -0.4074W[mm] + 302.6 \qquad (4)$$

with a root mean square error (RMSE) of 0.57 and coefficient of determination ($R^2$) of 0.9955. For ports 1 and 2, the central frequencies were set to 300 MHz and 280 MHz, respectively, and the patch antenna was excited at 550 MHz. Fig 6B illustrates the resonance frequency achieved after matching each port, showing that a selective frequency was successfully achieved.

## C. Simulation model and hybrid antenna array

In the array applicator with six modules, each bidirectional microstrip surface coil and patch antenna were tuned and matched. Fig 7A–7C show the tuning, matching, and coupling analysis with S1j between each element and patch antenna for the 280 MHz, 300 MHz and 550 MHz ports. The maximum coupling was -25 dB with a 280 MHz input source, -20 dB with a 300 MHz source, and -18 dB at 550 MHz for the patch antenna.

The coupling performance at 550 MHz was less critical since this frequency was used solely for heating production. Additionally, high-pass filters were implemented to further reduce coupling in the 280–300 MHz frequency band. Fig 7C shows that a high-pass filter can reduce the coupling to around -50 dB in the 280–300 MHz band. It should be noted that the proposed device is designed to operate at different times rather than simultaneously, meaning that heating with the patch antenna stops during the imaging of $^{19}$F and $^1$H.

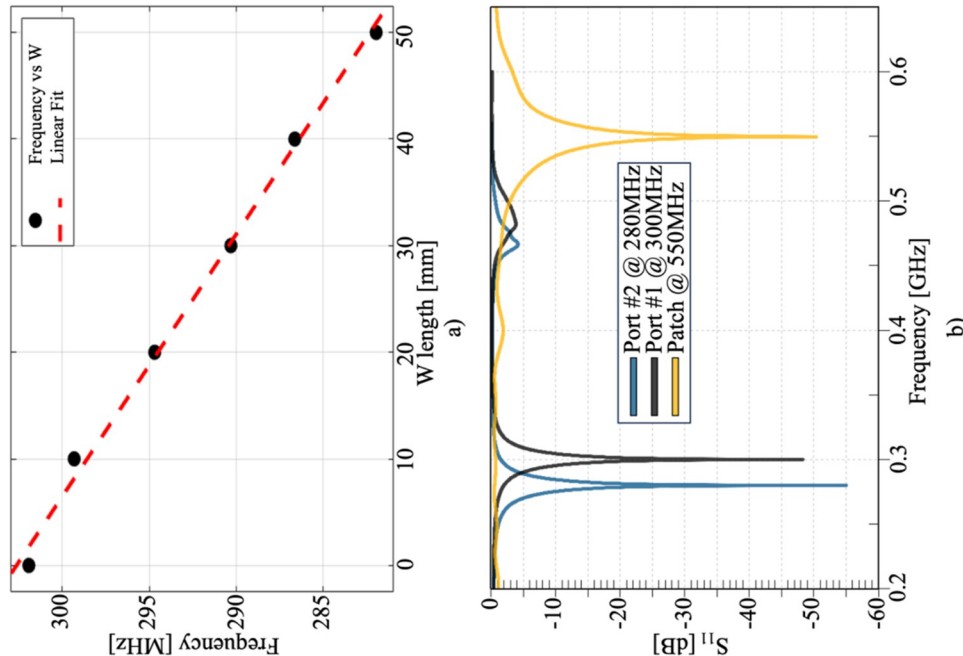

**Fig 6.** a) Analysis of the relationship between the length of W and the resonance frequency. b) Computed S11 parameter after tuning and matching for the single module, combining the bidirectional microstrip operating at 280 MHz and 300 MHz, and the patch antenna operating at 550 MHz.

Magnetic field maps of the $|B_1|$-field produced at 280 MHz and 300 MHz were computed using the simulation models (Fig 8A and 8B). The average $|B_1|$-field value for 280 MHz, applied to $^{19}$F, was 0.05 µT with a uniformity (NAAD) of 71.58%. For 300 MHz, the average $|B_1|$-field was 0.03 µT with a NAAD of 76.00%. The $|E|$-field of the patch antennas at 550 MHz

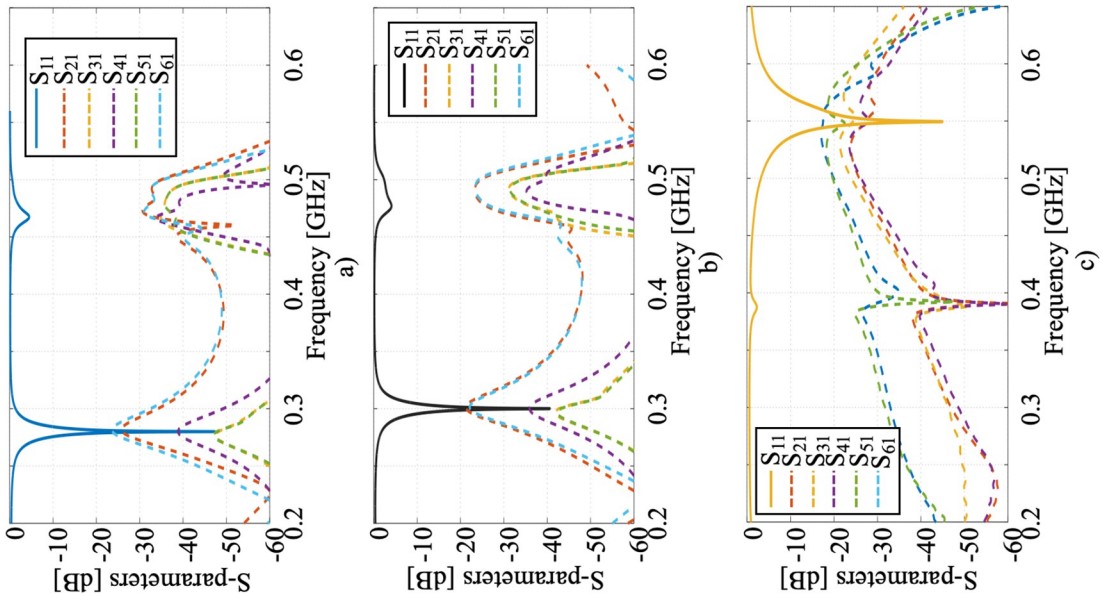

**Fig 7.** The plots show the scattering parameters S1j describing the coupling between each element, for the ports operating at the following frequencies: a) 280 MHz, b) 300 MHz, and c) 550 MHz.

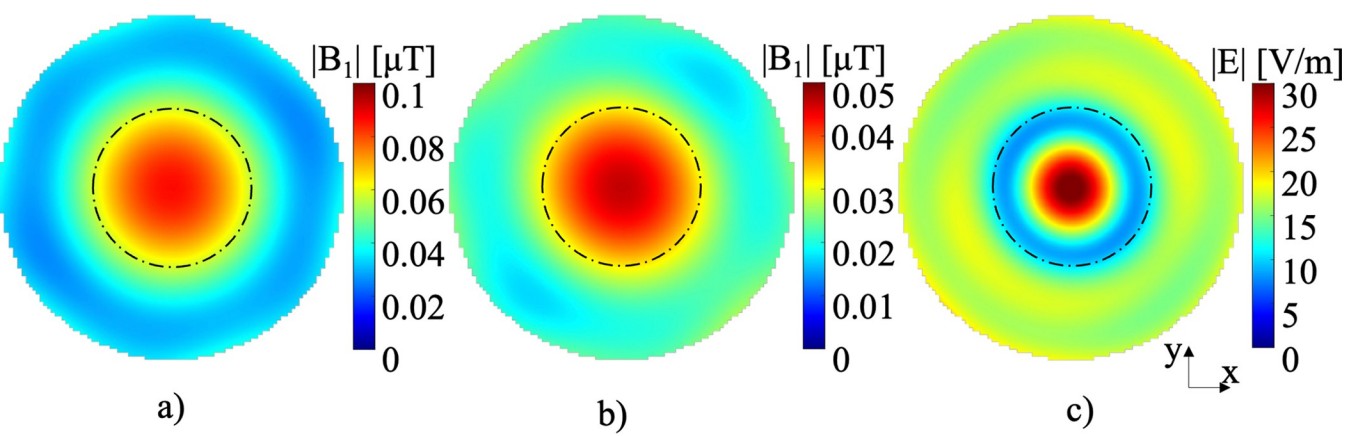

**Fig 8.** The magnetic $|B_1|$-field was computed with the array and simulation model for a) $^{19}$F at 280 MHz, and b) $^{1}$H at 300 MHz. c) The $|E|$-field at 550 MHz was calculated.

is shown in Fig 8C, with an average intensity of 15.92 V/m, and the field is localized at the center of the simulation model.

Fig 9 shows the temperature distributions resulting from the patch antenna array for three different cases: 1) the simulation model placed in the center, 2) shifted to the left, and 3) shifted to the upper left region. The simulation model was heated for 600 s. The hyperthermic metrics, including Tmax, T10, T50, and T90 within the cancer model, were measured at each time point. At 600 s, the Tmax, T10, T50, and T90 values for the three respective cases were approximately 43.20°C, 42.95°C, 42.40°C, and 40.87°C, respectively.

## D. Simulation study with a human model

**I) Performance of hyperthermia.** The distributions of $|B_1|$-fields at 280 MHz and 300 MHz are shown in Fig 10A and 10B, respectively. The average value and uniformity of the $|B_1|$-fields applied to $^{19}$F (280 MHz) were measured to be 0.21 μT and 81.75%, while for $^{1}$H (300 MHz), these values were 0.12 μT and 87.74%. The proposed antenna demonstrated a higher $|B_1|$-field intensity for the simulation model and human model with $^{19}$F, which was approximately twice that of $^{1}$H. This result is particularly promising for high-resolution $^{19}$F drug tracking, addressing the common challenge of lower signal intensity when utilizing X-nuclei in MRI. The $|E|$-field for HT heating, computed by the patch antennas at 550 MHz, is illustrated in Fig 10C, with an average value of 21.22 V/m in the cancer region.

Prolonged heating with a gradual temperature increase led to a steady-state HT temperature increase in the perfused cancer model. The temperature maps at 600 s for perfusion rates of 1 ml/min/kg and 3 ml/min/kg in the cancer are depicted in Fig 11A and 11C, respectively. Applied power levels of 120 W and 225 W were necessary to achieve similar maximum temperature increases in the cancer model. The time-variant temperature profiles within the cancer model are shown in Fig 11B and 11D, with the average temperature reaching approximately 40°C at 600 s.

**II) Safety evaluation: SAR distribution.** To report the safety performance of the hybrid antenna arrays within the MRI scanner, the SAR level was calculated for each frequency range, as shown in Fig 12. The $SAR_{10g}$ average and maximum for 280 MHz were measured to be 0.0198 and 0.0987 W/kg, respectively. At 300 MHz, the average and maximum SAR values were 0.0028 and 0.2969 W/kg, respectively. The computed SAR values at 550 MHz are shown in Fig 12C, with average and maximum SAR values of 0.0480 and 0.6088 W/kg, respectively.

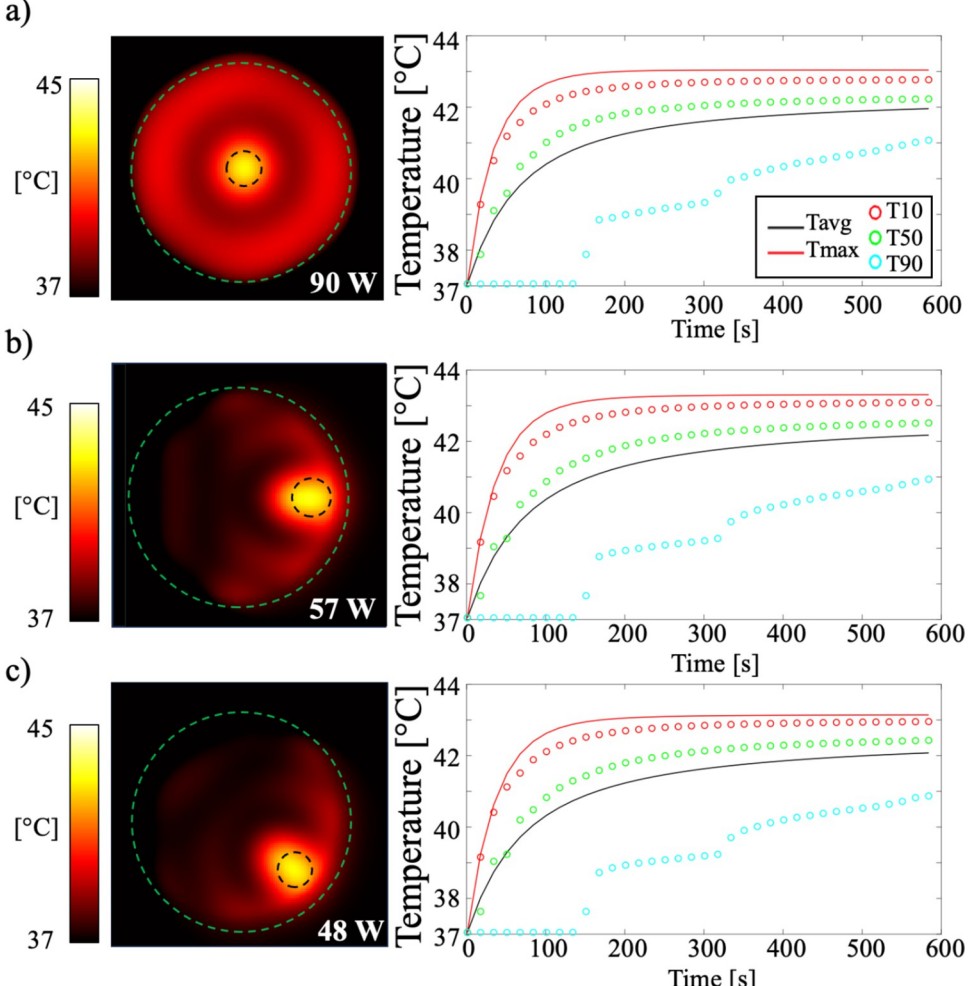

**Fig 9.** Temperature maps in the simulation model when placed at different positions: a) at the center, b) shifted to the left, and c) shifted to the upper left. Plots show the Tmax, T10, T50, and T90 values within the cancer model.

These results indicate that, with a normalization of 1W, the proposed antenna maintains safety levels below the limits specified in IEC 60601-2-33.

## IV. Discussion

In this study, we introduced a novel hybrid antenna array design capable of simultaneous operation at three distinct RF frequencies. This design facilitates the acquisition of images using multi-nuclei MR imaging techniques for both $^{19}$F and $^1$H nuclei, while also serving as a versatile tool for HT applications. The proposed applicator was specifically designed to image $^{19}$F-labeled drugs during HT along with $^1$H-based MR anatomical and thermal imaging.

Generally, the application of microstrips in MRI can be limited by their electrical length due to the operational frequency of MRI systems. For example, a quarter-wavelength in a 3T MRI system operating at 128 MHz corresponds to approximately 58 cm; in a 7T MRI system operating at 300 MHz, it is approximately 24 cm. These dimensions are impractical for MRI applications, necessitating innovative approaches to miniaturize the microstrips for MRI.

To overcome these limitations, we designed a bidirectional microstrip by introducing non-symmetric conductor paths (wings) that could operate at 280 MHz and 300 MHz, depending

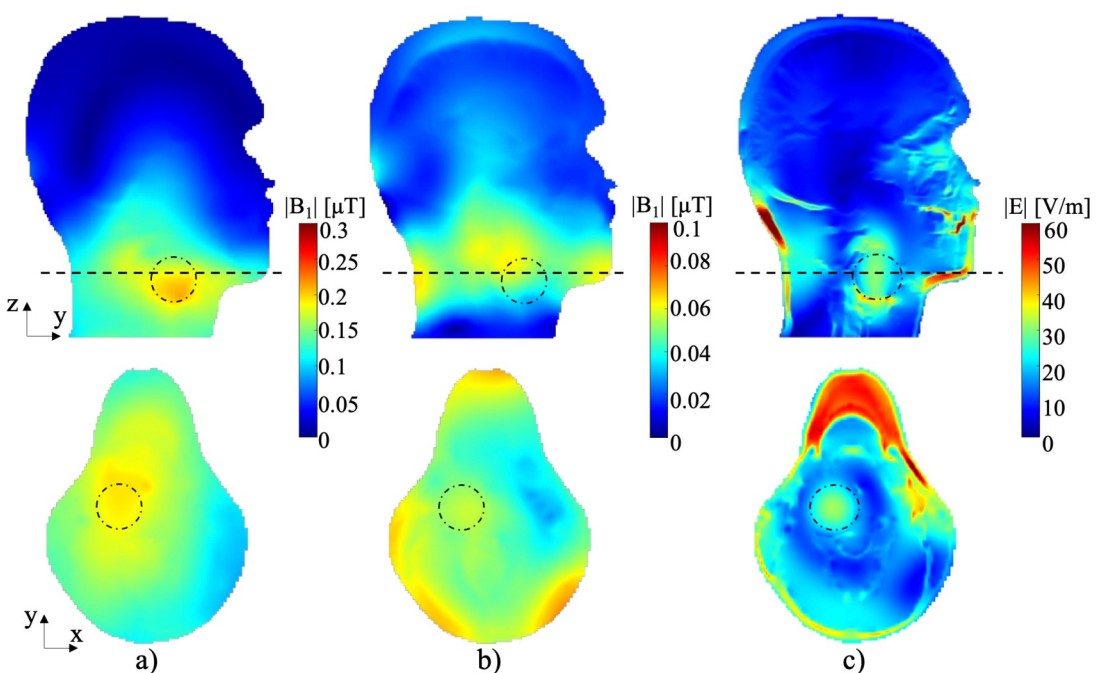

**Fig 10.** The computed |B₁|-field for a) ¹⁹F at 280 MHz, b) ¹H at 300 MHz, and c) electric field at 550 MHz on the Z-Y and X-Y planes within a human model.

on the excitation port, while fitting into a manageable size for neck applications in MRI. Additionally, the loop-microstrip design provided space to accommodate a patch antenna, thereby reducing concerns about coupling between coils or RF heating elements.

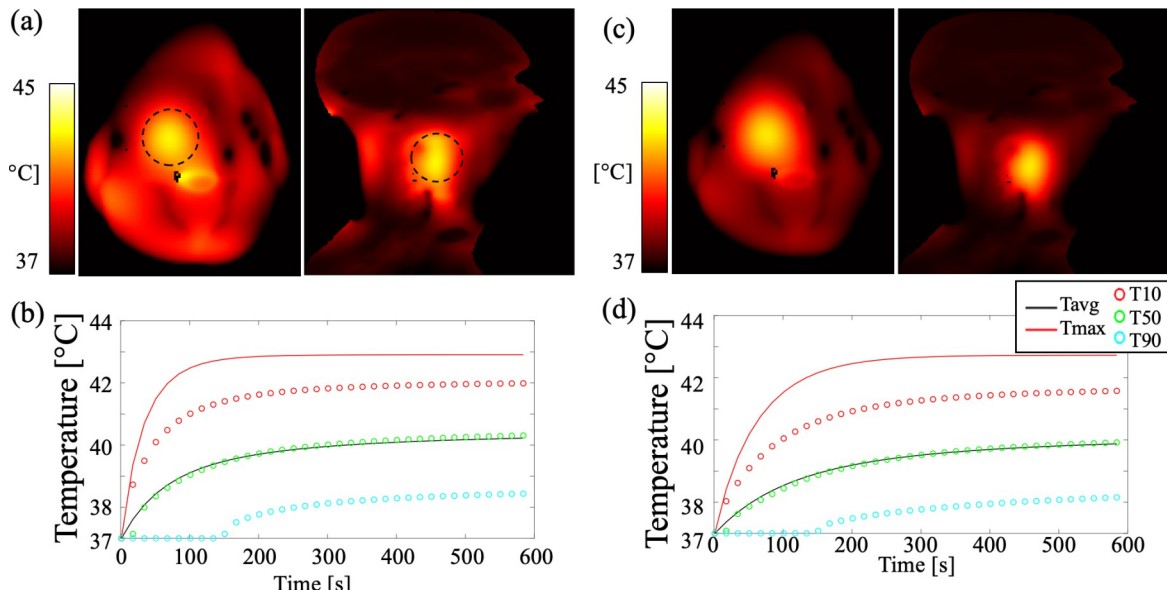

**Fig 11.** Temperature maps for perfusion rates of 1 ml/min/kg at 120 W (a) and 3 ml/min/kg at 225 W (c). The corresponding temperature profiles within the cancer model are shown in (b) and (d). The dashed black circles indicate the cancer model.

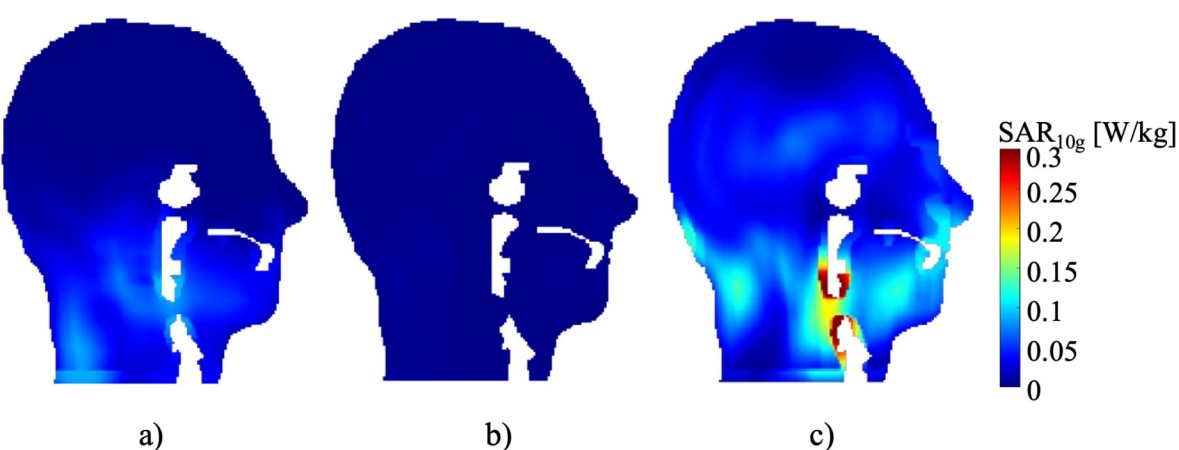

**Fig 12.** SAR maps on the head and neck with the proposed antenna for each frequency: a) 280 MHz, b) 300 MHz, and c) 550 MHz.

At 280 MHz for [19]F imaging, the $|B_1|$ field provides a higher field intensity within the cancer, enhancing signal detection because of the lower sensitivity of [19]F compared to that of [1]H. This phenomenon typically does not pose a concern for imaging, as the region of interest for [19]F imaging is confined to the cancer model [42]. Conversely, for [1]H imaging, achieving uniformity across the entire anatomical area of the neck is crucial for accurate anatomical imaging.

Another important aspect of the design is the selection of the water bolus surrounding the simulation model and the neck, as its electrical permittivity can significantly alter the entire E-field distribution. We observed that altering the permittivity of a water bolus could improve the focus of RF heating [43]. However, a detailed explanation of this effect is beyond the scope of this study and will be addressed in future research. In terms of RF safety, as evaluated by the SAR, the proposed coil provides an acceptable SAR value that is within the recommended safety limits.

In this study, six patch antennas were employed to generate localized heating for HT. The number of antennas was determined based on the size of the modules and the target organ. To enhance RF focusing, three key factors should be considered: 1) increasing the number of modules, 2) understanding the differences in electrical properties between tissues, and 3) considering the perfusion rates of cancer and surrounding tissue. The E-field patterns can be influenced by the electrical properties of cancerous tissues. In this study, the electrical properties of the cancer model were selected based on previous studies, and localized heating was achieved using six array antennas [44, 45]. However, further research is needed to optimize the number of modules for other types of cancer, particularly where there is little contrast in the electrical properties between cancerous and healthy tissues.

Table 2 summarizes a comparison of the main characteristics of the proposed microstrip design and similar works found in the literature.

**Table 2. Comparison between commercial and/or research MR-guided microwave HT systems.**

| Reference | Nuclei | MRI scanner | Target organ | Antenna type | Frequency [MHz] |
|---|---|---|---|---|---|
| Pyrexar BSD2000-3D | [1]H | 1.5T | Body | 12 pairs of dipole antenna | 75–120 |
| Sumser et al [29] | [1]H | 1.5T | Head and neck | 12 channel Yagi-Uda | 433.92 |
| Winter et al [28] | [1]H | 7T | Brain | 8 channel Bow-tie | 298 |
| Our study | [1]H, [19]F | 7T | Head and neck | 6 channel microstrip | 550 |

In clinical practice, the perfusion rate of cancer tissues can significantly affect the effectiveness of HT. Our simulation results with the human model indicate that the electrical power required to achieve the desired HT temperature range varies accordingly. Future studies should focus on optimizing the power levels based on the estimated perfusion rates and specific target locations.

A limitation of our study is that the thermal simulations were conducted without temperature feedback control to maintain HT heating. Although we simulated prolonged heating with a slow temperature elevation to approximate mild HT heating with a steady-state temperature of 43˚C, future research should investigate HT distributions with feedback control, considering the influence of tissue perfusion over extended periods. Another limitation is that the fabrication of the applicator and the experimental validation were not demonstrated. The construction of an antenna is a challenging task that requires precise attention to the dimensions of both the conductor and dielectric materials, as the operational frequency of the device depends on the length of the conductor. Ensuring the correct frequency and loading conditions adds to complexity. Subsequent investigations will focus on validating the device fabrication process, demonstrating its efficacy for imaging with [19]F-based contrast agents during HT procedures, and evaluating the $|B_1|$-field intensity and uniformity using MRI-based field mapping techniques.

## V. Conclusion

This study introduces a new design for a versatile RF applicator using a hybrid antenna array that enables [1]H proton imaging, [19]F drug tracking, and HT. We presented a loop-type bidirectional microstrip design to produce uniform and strong $|B_1|$-fields for both [19]F and [1]H, and integrated it with patch antennas for RF focusing to heat cancer located in the neck. Simulation studies using a human model demonstrated the feasibility of HT heating and multi-nucleus MR imaging. Although the applicator was specifically designed for neck HT treatment, the design and techniques used in this study could potentially be adapted for brain tumor treatment in the future.

## Author Contributions

**Conceptualization:** Daniel Hernandez, Jae Jun Lee, Kisoo Kim, Kyoung Nam Kim.

**Data curation:** Daniel Hernandez, Taewoo Nam, Eunwoo Lee.

**Formal analysis:** Daniel Hernandez, Taewoo Nam, Eunwoo Lee.

**Funding acquisition:** Jae Jun Lee, Kisoo Kim, Kyoung Nam Kim.

**Investigation:** Daniel Hernandez, Taewoo Nam.

**Methodology:** Daniel Hernandez, Taewoo Nam, Eunwoo Lee, Kisoo Kim, Kyoung Nam Kim.

**Project administration:** Jae Jun Lee, Kyoung Nam Kim.

**Resources:** Jae Jun Lee, Kyoung Nam Kim.

**Software:** Daniel Hernandez, Taewoo Nam, Eunwoo Lee.

**Supervision:** Jae Jun Lee, Kisoo Kim, Kyoung Nam Kim.

**Validation:** Daniel Hernandez.

**Visualization:** Daniel Hernandez, Eunwoo Lee, Kisoo Kim.

**Writing – original draft:** Daniel Hernandez, Taewoo Nam, Eunwoo Lee, Jae Jun Lee, Kisoo Kim, Kyoung Nam Kim.

**Writing – review & editing:** Daniel Hernandez, Jae Jun Lee, Kisoo Kim, Kyoung Nam Kim.

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
