## [Decision Letter · Decision Letter 0]

2 Aug 2024

PONE-D-24-21640Computational Design of Multi-modal Antenna Arrays for Microwave Hyperthermia, 1H MR Thermometry, and 19F Drug Release MRI MonitoringPLOS ONE

Dear Dr. Kim,

Thank you for submitting your manuscript to PLOS ONE. After careful consideration, we feel that it has merit but does not fully meet PLOS ONE’s publication criteria as it currently stands. Therefore, we invite you to submit a revised version of the manuscript that addresses the points raised during the review process.

We look forward to receiving your revised manuscript.

Kind regards,

Mai Osama Mohamed Ibrahim Sallam

Academic Editor

PLOS ONE

Journal Requirements:

3. Thank you for stating the following financial disclosure: "1. National Institutes of Dental and Craniofacial Research (NIDCR), USA under Grant K99DE032397

2. Osong Medical innovation Foundation R and D Project funded by the Republic of Korea’s Ministry of Health and Welfare (No: HI22C1989)"

4. We note that your Data Availability Statement is currently as follows: "All relevant data are within the manuscript and its Supporting Information files."

Reviewers' comments:

Reviewer's Responses to Questions

**Comments to the Author**

1. Is the manuscript technically sound, and do the data support the conclusions?

Reviewer #1: Partly

Reviewer #2: Yes

2. Has the statistical analysis been performed appropriately and rigorously? 

Reviewer #1: N/A

Reviewer #2: Yes

3. Have the authors made all data underlying the findings in their manuscript fully available?

Reviewer #1: No

Reviewer #2: Yes

4. Is the manuscript presented in an intelligible fashion and written in standard English?

Reviewer #1: No

Reviewer #2: No

5. Review Comments to the Author

Reviewer #1: Daniel and colleagues present the design and numerical analysis of a multi-modal antenna for MW Hyperthermia and MRI, which is a sound topic. However, corrections must be included in the manuscript.

Minor comments:

(1) Review typos, missing spaces, and minor format issues.

(2) The diagrams in Figures 1 to 3 are difficult to read. Please change the colors, check the typos and values, and clarify the antenna design.

(3) Is there a repository where the data is available?

Major comments:

(1) Drug release MRI monitoring is an adequate application for the proposed antenna, exposed in the introduction. However, follow-up has not actually been studied. Monitoring of evolving clinical conditions should be added or removed from the title. In another way, the title and abstract are misleading.

(2) The topic of MW hyperthermia is well-studied, and the art could be better referenced. Moreover, aspects such as the water interphase (matching medium), the number of antennas, and its dimensions are relevant to the article and need to be further discussed.

(3) First, the Phantom word gives an idea of the ones used experimentally. I suggest using a term like an anthropomorphic numerical model. Second, considering it is a numerical study, why don't we use a realistic model of cancer or a more representative one in terms of location and shape from a medical point of view?

(4) This is a preliminary study of an antenna design and is thus far from a feasibility study. Please adjust the stated scope of the manuscript.

Reviewer #2: This paper presents a design for an antenna system to be used in Microwave Hyperthermia. The first comment I have is on the title:

- "Computational Design of Multi-modal Antenna Arrays for Microwave Hyperthermia, 1H MR Thermometry, and 19F Drug Release MRI Monitoring"

- The word "Computational" seems to be unnecessary because most design in modern research is supposedly computational. Analytical designs are very rare.

Looking at the abstract, the writing of the abstract is poor and long (267 words):

- "This device enables microwave hyperthermia for releasing 19F-labeled drugs from thermosensitive liposomes, monitoring of 19F-labeled drugs, and 1H MR thermometry within a 7T MRI system", this sentence is vague and doesn't seem to match the title. The sentence states that microwave hyperthermia is a medium for 19F-labeled drugs release and Thermometry. However, the title portrays that "Microwave Hyperthermia" is a separate application from Thermometry and drug release. Second, the "7T MRI system" place in the sentence is vague as it doesn't make it clear if the "7T MRI system" is only related to "1H MR thermometry" or related to the whole sentence.

- "Electromagnetic and thermal simulations were performed to evaluate the performance of the applicator with a phantom and a human model to demonstrate the feasibility of hybrid antenna array for clinical translation", unnecessary sentence because the later sentences make it obvious that electromagnetic and thermal simulations were performed.

- "A hybrid antenna array applicator was developed to perform 1H MR imaging, MRI 19F-drug tracking, and hyperthermia. A new concept of loop-type bidirectional microstrip allowed generating uniform |B1|-fields for 19F and 1H and integrating with patch antennas for hyperthermia.", these sentences are redundant.

As for the introduction, the authors jump directly into talking about chemotherapy without a first general introductory paragraph on cancer treatment in general and what microwave hyperthermia has to offer, and then start expanding this first paragraph by talking about each treatment method. Also why is there no paragraph talking about B1 and |E| fields and why the B1 fields were chosen

There are a lot of grammatical mistakes and typos. Please run through Grammarly. There are also capitalizing errors such as writing "table" instead of "Table".

The author must illustrate the novelty of the study

Please add a comparison table to compare your work with the literature.

6. PLOS authors have the option to publish the peer review history of their article (what does this mean?). If published, this will include your full peer review and any attached files.

Reviewer #1: No

Reviewer #2: No

---

## [Author Response · Author response to Decision Letter 0]

28 Aug 2024

PONE-D-24-21640

Computational Design of Multi-modal Antenna Arrays for Microwave Hyperthermia, 1H MR Thermometry, and 19F Drug Release MRI Monitoring

PLOS ONE

Comments to the Author

Reviewer #1: Daniel and colleagues present the design and numerical analysis of a multi-modal antenna for MW Hyperthermia and MRI, which is a sound topic. However, corrections must be included in the manuscript.

Minor comments:

(1) Review typos, missing spaces, and minor format issues.

Ans: We have revised the document and corrected typos and format mistakes throughout the manuscript. 

(2) The diagrams in Figures 1 to 3 are difficult to read. Please change the colors, check the typos and values, and clarify the antenna design.

Ans: We have updated the figures for better visualization, and we corrected the typos in the figures.

(3) Is there a repository where the data is available?

Ans: As suggested, we have deeply considered sharing data and 3D models we developed in this study. We will use the ‘DRYAD repository to share STL models we used and documents for simulation parameters. We will complete the repository submission and upload all STL models and simulation instruction when the manuscript is accepted and published.

Major comments:

(1) Drug release MRI monitoring is an adequate application for the proposed antenna, exposed in the introduction. However, follow-up has not actually been studied. Monitoring of evolving clinical conditions should be added or removed from the title. In another way, the title and abstract are misleading.

Ans: We agree with the reviewer’s suggestion; therefore, we have changed the title to “Design of Multi-modal Antenna Arrays for Microwave Hyperthermia and ¹H/¹⁹F MRI Monitoring of Drug Release,” which indicates that it can be used as an application, and we have also modified the abstract to indicate the real scope of our work.

(2) The topic of MW hyperthermia is well-studied, and the art could be better referenced. Moreover, aspects such as the water interphase (matching medium), the number of antennas, and its dimensions are relevant to the article and need to be further discussed.

Ans: We appreciate the reviewer’s comment and have revised the documents and updated them with the information suggested in the Methods section. In addition, in the discussion section on the water bolus (matching medium), we added a reference that addresses this study.

(3) First, the Phantom word gives an idea of the ones used experimentally. I suggest using a term like an anthropomorphic numerical model. 

Ans: Thank you for bringing this point to our attention. We agree with reviewer’s comment but we thought the word "anthropomorphic" is not commonly used in related works as well. Hence, we would suggest using the word ‘simulation model.’ 

Second, considering it is a numerical study, why don't we use a realistic model of cancer or a more representative one in terms of location and shape from a medical point of view?

Ans: We considered using a more realistic or representative cancer model in terms of location and shape. In our study, we opted to use a spherical tumor model as a simplified representation to focus on the fundamental aspects of the simulation, such as the computational feasibility and parameters. Although we acknowledge that this model does not capture the full complexity of real tumor shapes, it allows us to systematically investigate the key dynamics and interactions within the human model. A spherical tumor serves as a baseline that can be generalized or extended in future studies. We plan to explore more complex and realistic tumor geometries, informed by medical imaging data, in subsequent research to validate and expand the findings presented here.

(4) This is a preliminary study of an antenna design and is thus far from a feasibility study. Please adjust the stated scope of the manuscript.

Response: Following the reviewer’s suggestion, we have modified the Abstract, Introduction, and Discussion sections to indicate that this study is based on a preliminary study based on simulations.

Reviewer #2: This paper presents a design for an antenna system to be used in Microwave Hyperthermia. The first comment I have is on the title:

- "Computational Design of Multi-modal Antenna Arrays for Microwave Hyperthermia, 1H MR Thermometry, and 19F Drug Release MRI Monitoring"

- The word "Computational" seems to be unnecessary because most design in modern research is supposedly computational. Analytical designs are very rare.

Ans: Following the reviewer suggestion we have removed the word computational from the title, and we have updated the title to: Design of Multi-modal Antenna Arrays for Microwave Hyperthermia and ¹H/¹⁹F MRI Monitoring of Drug Release.

Looking at the abstract, the writing of the abstract is poor and long (267 words):

- "This device enables microwave hyperthermia for releasing 19F-labeled drugs from thermosensitive liposomes, monitoring of 19F-labeled drugs, and 1H MR thermometry within a 7T MRI system", this sentence is vague and doesn't seem to match the title. The sentence states that microwave hyperthermia is a medium for 19F-labeled drugs release and Thermometry. However, the title portrays that "Microwave Hyperthermia" is a separate application from Thermometry and drug release. Second, the "7T MRI system" place in the sentence is vague as it doesn't make it clear if the "7T MRI system" is only related to "1H MR thermometry" or related to the whole sentence.

Ans: We appreciate the reviewer comment, we have revised the abstract by reducing the word count within the limits, we have improved the meaning of the paragraph and changed to: The device aims to provide microwave hyperthermia to release 19F-labeled cancer treatment drugs from thermosensitive liposomes. This approach allows for monitoring the concentration of these drugs through 19F imaging and enables 1H MR thermometry for temperature control. The design consists of a bidirectional microstrip to generate magnetic |B1|-fields at 300 MHz and 280 MHz for 1H and 19F MR imaging, respectively, and a patch antenna to produce localized RF heating for hyperthermia.”

The proposed device is designed to fulfill three key functions:

1. Microwave Hyperthermia: This function is intended for both therapeutic heating and drug delivery. The device facilitates microwave hyperthermia to deliver localized heating, which is essential for activating cancer treatment drugs that are tagged with 19F. These drugs are engineered to release their payload at specific temperatures. The inclusion of the patch antenna helps achieve precise, focused heating.

2. 19F Imaging: Since the anticancer drugs are labeled with 19F, monitoring their delivery and distribution within the body is crucial. The proposed microstrip antenna is designed to acquire 19F images using a 7T MRI scanner, enabling detailed tracking of the drug's distribution and effectiveness.

3. 1H Imaging: The microstrip antenna also supports 1H imaging, which can be used for anatomical imaging or temperature monitoring (thermometry). This capability complements the 19F imaging by providing additional information on the body's structure and temperature changes.

- "Electromagnetic and thermal simulations were performed to evaluate the performance of the applicator with a phantom and a human model to demonstrate the feasibility of hybrid antenna array for clinical translation", unnecessary sentence because the later sentences make it obvious that electromagnetic and thermal simulations were performed.

Ans: We have updated the sentence to remove the unnecessary words, the new sentence is : “Simulation results using a tissue-mimicking phantom confirmed the intensity and uniformity of |B1|-fields for both 19F and 1H nuclei, demonstrating the suitability of the design for clinical imaging. The RF heating from the patch antennas was effectively localized at the center of the cancer model.”

- "A hybrid antenna array applicator was developed to perform 1H MR imaging, MRI 19F-drug tracking, and hyperthermia. A new concept of loop-type bidirectional microstrip allowed generating uniform |B1|-fields for 19F and 1H and integrating with patch antennas for hyperthermia.", these sentences are redundant.

Ans: We have removed the sentence and it was updated as : “This study demonstrates the hybrid antenna array's potential for integrating 1H MR imaging, 19F drug monitoring, and hyperthermia in advanced cancer treatment.”

As for the introduction, the authors jump directly into talking about chemotherapy without a first general introductory paragraph on cancer treatment in general and what microwave hyperthermia has to offer, and then start expanding this first paragraph by talking about each treatment method. Also why is there no paragraph talking about B1 and |E| fields and why the B1 fields were chosen

Ans: Following the reviewer’s suggestion, we have modified the introduction to discussion the importance of cancer treatments. The paragraphs now read as: “Head and neck cancer is one of the most commonly diagnosed cancers, with increasing incidence and mortality rates [1]. Effective treatment is crucial for survival. Current treatment options include surgery, radiation therapy, and chemotherapy. Although chemotherapy is effective, systemic treatment can cause severe toxic side effects on normal cells and organs. To address these challenges, targeted drug delivery technologies have been developed to enhance treatment efficacy by increasing the concentration of drugs at the tumor site while minimizing exposure to healthy tissues. These technologies achieve this by targeting drugs to specific cells, releasing drugs in response to certain conditions, or using heat therapy to promote drug release from liposomes [2–6].”

We have modified the introduction to emphasize the importance of the |B1|-field for MR imaging and the |E|-field for hyperthermia applications: “For MRI, it is essential that coils or antennas produce uniform and strong magnetic |B1|-fields, which are crucial for both signal transmission and reception [29]. These |B1|-fields must operate at frequencies corresponding to the Larmor frequency of each nucleus being imaged. The Larmor frequency is determined by the strength of the main magnetic field (B0) of the MRI scanner. For instance, in a 7T MRI scanner, the Larmor frequency for 1H is approximately 300 MHz, while for 19F it is around 280 MHz. Ensuring that the coils or antennas produce |B1|-fields at these specific frequencies is vital for obtaining high-quality images and accurate data during MRI procedures.

In MRI, the primary characteristic of the antenna is its ability to generate the |B1|-field. It is also desirable for the electric |E|-field to be minimal to reduce the specific absorption rate (SAR), a key safety concern in MRI. However, for the development of thermal applicators, the focus shifts to enhancing |E|-field generation. A higher |E|-field interacts with tissues, and through Ohmic resistance, it absorbs energy that can raise tissue temperature. Consequently, patch antennas, dipole antennas, and bowtie antennas are commonly used for non-invasive temperature generation in medical applications due to their effective |E|-field generation capabilities. With the |E|-field temperature maps can be calculated based on the Pennes bioheat method [30].”

There are a lot of grammatical mistakes and typos. Please run through Grammarly. There are also capitalizing errors such as writing "table" instead of "Table".

Ans: We have revised the document and fixed the typos and format mistakes.

The author must illustrate the novelty of the study

Ans: We have modified the introduction to show the novelty of our study: “The aim of this study was to develop a novel hybrid antenna array that integrates three key functions: microwave hyperthermia, and 19F and 1H imaging. We introduce the concept of a bidirectional microstrip antenna that creates asymmetric current pathways along the conductor line to produce two different frequencies seen from the excitation ports at both ends of the antenna. To ensure compatibility with the MRI and human size scales, the bidirectional microstrip antenna was modified to reduce size by implementing the conductor line in a loop shape while maintaining asymmetrical current pathways between the ports. This design also includes a centrally located patch antenna within the loop microstrip to deliver targeted heat for hyperthermia. The proposed microstrip and patch antenna arrangement was extended into an array setup. Electromagnetic and thermal simulations were performed to calculate |B1|-fields, |E|-fields, and temperature maps for a neck cancer model, demonstrating the feasibility of the proposed antenna array for clinical translation.”

Please add a comparison table to compare your work with the literature.

Ans: Thank you for your suggestion. As recommended, we have created a comparison table of related studies and included it in the discussion section.

---

## [Decision Letter · Decision Letter 1]

7 Oct 2024

Design of Multi-modal Antenna Arrays for Microwave Hyperthermia and ¹H/¹⁹F MRI Monitoring of Drug Release

PONE-D-24-21640R1

Dear Dr. Kim,

We’re pleased to inform you that your manuscript has been judged scientifically suitable for publication and will be formally accepted for publication once it meets all outstanding technical requirements.

Kind regards,

Mai Osama Mohamed Ibrahim Sallam

Academic Editor

PLOS ONE

Additional Editor Comments (optional):

Reviewers' comments:

Reviewer's Responses to Questions

**Comments to the Author**

1. If the authors have adequately addressed your comments raised in a previous round of review and you feel that this manuscript is now acceptable for publication, you may indicate that here to bypass the “Comments to the Author” section, enter your conflict of interest statement in the “Confidential to Editor” section, and submit your "Accept" recommendation.

Reviewer #1: All comments have been addressed

Reviewer #2: All comments have been addressed

2. Is the manuscript technically sound, and do the data support the conclusions?

Reviewer #1: Yes

Reviewer #2: Yes

3. Has the statistical analysis been performed appropriately and rigorously? 

Reviewer #1: N/A

Reviewer #2: Yes

4. Have the authors made all data underlying the findings in their manuscript fully available?

Reviewer #1: Yes

Reviewer #2: Yes

5. Is the manuscript presented in an intelligible fashion and written in standard English?

Reviewer #1: Yes

Reviewer #2: Yes

6. Review Comments to the Author

Reviewer #1: Thank the authors for addressing all the comments and suggestions. The manuscript has improved and is now more readable and clear. Last minor comment: be careful with the resolution of the images during the editorial part.

Reviewer #2: the authors have addressed the required comments, and I think the paper is ready for publication in PLOS one journal

7. PLOS authors have the option to publish the peer review history of their article (what does this mean?). If published, this will include your full peer review and any attached files.

Reviewer #1: No

Reviewer #2: No

---

## [Editor Report · Acceptance letter]

10 Oct 2024

PONE-D-24-21640R1 

PLOS ONE

Dear Dr. Kim, 

I'm pleased to inform you that your manuscript has been deemed suitable for publication in PLOS ONE. Congratulations! Your manuscript is now being handed over to our production team.

Kind regards, 

on behalf of

Dr. Mai Osama Mohamed Ibrahim Sallam 

Academic Editor

PLOS ONE